# Towards Sustainable Collaborative Logistics Using Specialist Planning Algorithms and a Gain-Sharing Business Model: A UK Case Study

**Alix Vargas** [1,*] **, Carmen Fuster** [1] **and David Corne** [2]

1   Connected Places Catapult (CPC), Milton Keynes MK9 1BP, UK; carmen.Fuster@cp.catapult.org.uk
2   School of Mathematical & Computer Sciences, Heriot-Watt University, Edinburgh EH14 4AS, UK;
    D.W.Corne@hw.ac.uk
*   Correspondence: a.vargas@herts.ac.uk; Tel.: +44-7437343123

**Abstract:** This paper introduces the FreightShare Lab Platform (FSLP) and its embedded business model, aiming to facilitate and encourage horizontal collaboration in freight logistics. The idea of the FSLP is to create collaborating clusters of freight operators, and corresponding collaborative operational plans, via specialised decision support algorithms and multi-fleet optimisation. Further, a gain-sharing business model embedded within the FSLP algorithms ensures that participants, mainly logistics service providers and freight operators, can retain their own profit margins and fairly share the efficiency gains from collaboration. A case study is presented, centred on a large UK freight operator, to evaluate the key FSLP algorithms in a realistic context. The results evidence the potential for significant financial and environmental benefits for industry and society.

**Keywords:** business model; gain sharing; planning algorithms; logistics collaboration; physical internet

## 1. Introduction

Horizontal collaboration in the logistics industry has the potential to generate sustainable benefits for society, the environment and the economy. Additionally, logistics service providers would benefit from reduced operating costs resulting from the use of fewer trucks, lower mileage and increased fleet utilisation, from which—assuming perfect competition—customers would also benefit [1]. According to the World Business Council for Sustainable Development, collaboration between logistics operators using freight exchanges can yield cost savings of c.20% and a reduction in $CO_2$ emissions of c.32% [2]. Collaboration is also a key cornerstone of developments towards the creation of a Physical Internet.

Even when the theory shows evidence of benefits for industry and society using collaboration in the logistics industry, the reality is that the market is not ready for such a change. The barriers for sharing logistics assets (trucks and/or jobs) with others (potential competitors) and for collaborating are significant. Among them are: a lack of trust between potential collaborators, a refusal to share data and a reluctance to change a business model that is based on competing for market share [1]. These establish the foundations of the problem that needs to be overcome.

Some may be sceptical that collaboration can be consistent with the principles of competition and in a market in which participants compete for market share. The FreightShare Lab Platform (FSLP) project confronts such scepticism by showing that so-called "coopetition" can be established in a horizontal collaboration platform, demonstrating that elusive 'win-win' business case [1]. The FSLP does this by bridging the gap between the organisational theory of collaboration and the need for ICTs that understand the foundations of collaborative networks and provides the tools that enable collaboration in traditional road freight operations, supported by a win-win business model. The FSLP

aims to increase competitiveness, efficiency and the utilisation of assets between participating freight operators by creating a collaborative ecosystem.

The FSLP embeds the collaboration process by using two algorithms: The first algorithm facilitates the creation of clusters based around operational compatibility, and the second is a feature-rich fleet planning and optimization algorithm based on an operating commercial planning platform, which has been adapted to handle additional features related to inter-fleet sharing and a gain-share business model. The latter algorithm seeks a joint itinerary that can fulfil contracts at a lower price and with lower emissions by exploiting the combination of assets available to the FSLP [1]. The business model ensures that both the original Contract Holder (CH) (The freight operator which inputs the specific job contracted through carrier-shipper relationship outside FSLP, into FSLP, but does not necessarily fulfil the job) and those companies that provide the assets deployed by the FSLP to fulfil the contractor, called in this model the Contract Performer (CP) (The freight operator which inputs their resources available into FSLP and receives the routing and scheduling under which to deliver the job) have their fulfilment operating costs covered and retain their own profit margins [1]. In addition to that, they also might share the gains of the collaboration, based on the differential between their operating costs, estimated by the FSLP.

Sustainable societal and environmental benefits also result from collaboration through the FSLP. The second algorithm will always seek options that deliver emissions that are lower than if the CH had fulfilled the operation alone. The platform therefore rewards those operators that submit their contracts to the FSLP and those that the FSLP algorithm determines are the 'best' to fulfil the contract (i.e., based on the above cost-price-emissions criteria). This feature drives competition between the participants in the FSLP. The FSLP connects logistics providers with a wider pool of asset owners and operators: The larger the pool, the more chance there is of finding a better solution. In theory, the companies participating can only gain financially from this model [1].

The initial results obtained from model simulations using realistic data indicate that significant financial benefits exist for FSLP members using this 'gain-sharing' model [1]. The analysis, further using a UK case study, demonstrates evident reductions in the total mileage travelled, thereby indicating increases in the average utilisation of trucks, therefore heading to decreased road congestion and emissions.

This paper is structured in the following way: Section 2 presents the foundations of the literature review. Section 3 presents the novel Gain-Sharing Business Model. Section 4 presents the logic behind the Special Planning Algorithms for Logistics Collaboration. Section 5 presents the Integration of the Business Model and the FSLP Algorithms. Section 6 presents a UK Case Study which demonstrates the benefits of collaboration and suggests wider sustainable socio-economic and environmental advantages. Section 7 provides conclusions and suggests further work.

## 2. Background

Collaboration in the supply chain has been discussed and applied extensively in both industry and academic circles [3,4]. A number of different organisations are already collaborating in their Supply Chains (SC) to lower their operating costs, improve efficiency in the chain of operations and increase customer satisfaction through the sharing of information and assets, and through better coordinated network activities [5], thereby generating synergistic advantages that companies cannot achieve independently.

According to [6], "it is important to have a driving motive for all parties to work together, becoming a 'committee of equals' that finds greater value in collaboration to ensure long-term success". Additionally, allowing coordination among parties will facilitate the meeting of common business objectives [7]. Collaboration is possible when at least two actors share their efforts, data and/or assets to achieve mutual goals [8].

An increasing number of diverse Collaborative Networks (CN) have emerged in other industries because of advances in ICT, market and societal needs, and the progress made in many international projects [9].

*2.1. Collaboration in Freight Logistics*

The scarcity of collaborative initiatives in freight logistics is mainly due to competition between fleet operators and the sector's relatively low profit margins [10]. A growing number of small or medium-sized carriers have launched CNs in a bid to increase profit margins and competitiveness, yet significant sector inefficiencies still remain [11].

The freight logistics industry drives and enables economic growth and is a significant employer in Europe [12], but it also imposes significant societal costs [13]. These include health and environmental costs from pollution and traffic congestion, delays borne by road users and nuisance costs such as noise [1].

The UK's Department for Transport (DfT) has stated that 'empty running' in the UK rose from 27% to 30% between 2006 and 2016 [14]; in addition to that, capacity utilisation is only 68%, and freight efficiency is just 47.6%. Trucks are 'on the road' for barely a third of their time; the remaining two-thirds being idle periods including driver resting times and weekends, etc. All these antecedents result in an asset efficiency of only 15% to 16% [15]. Collaboration would reduce the number of HGVs on the motorway, decrease emissions, reduce empty running, and identify routes and journeys where operators can consolidate their loads into single vehicle trips [16]. With the right business models, opportunities for further collaboration abound.

2.1.1. Collaboration: Forms and Strategies

In [1], two different, but inter-linked, collaborative approaches were identified. The first defines who takes part in the collaboration, therefore distinguishes its physical structure. Three main categories apply specifically for the transport industry [17,18]:

1.　Vertical collaboration, which concerns two or more organisations at different levels of the logistics chain.
2.　Horizontal collaboration, which concerns two or more competing organisations at the same level of the logistics network.
3.　Multilateral collaboration, which combines and sharing capabilities both vertically and horizontally.

The second approach identifies the different types of coordination structure established between the participants [19,20]:

1.　Centralised, involving decision-making at a common higher level by generating synchronized instructions at lower levels
2.　Decentralised, implying a consensus, agreement of objectives, indicators and equality rules between partners. This collaboration is usually achieved through communication and negotiation between the partners.

Collaboration can include the following activities: transport of goods, warehousing, equipment pooling (e.g., container pools, pallet networks, etc.) and other operations. Collaboration usually takes place in the form of agreements and partnerships among a small number of companies and may even be ad-hoc rather than formal arrangements [21]. Different authors have identified some strategies of collaboration [11,16,22], e.g., cooperative alliances, route scheduling/planning, backhauling, freight exchanges, consolidation centres, delivery and servicing plans, and joint optimisation of assets and sharing capacity.

### 2.1.2. Collaboration: Barriers and Enablers

The work undertaken on [10] compiled the main barriers and limitations found in the literature, and strategies to overcome this are listed in Table 1.

**Table 1.** Barriers for collaboration in the freight industry and strategies to overcome them.

| Barriers/Limitations for Collaboration | Author | Strategies to Overcome Them |
|---|---|---|
| (A) Shipper concerns of having a different carrier from its usual contracted carrier. | [11] | Concerns over branding could be resolved through use of independent third parties and non-liveried vehicles. Involving the shipper into the alliance, through agreements, showing them the advantages of collaboration. |
| (B) Load compatibility can restrict the ability for loads to be shared. | [16] | Matching companies moving similar products with similar handling equipment on similar types of vehicles. |
| (C) Responsibility for transportation operations. | [23] | If the collaborations for logistics sharing follow a contract or a chart where the responsibilities are well defined, these questions will not constitute an obstacle to sharing. |
| (D) Legal barriers; there are laws that interfere with the ability to share data: competition law. | [16,23–26] | The European Union (EU) recommends the use of a neutral trustee, to whom different stakeholders give data to be held and analysed preventing the transfer of commercial data such as volumes, delivery addresses, costs, product characteristics, etc. |
| (E) Lack of human resources, especially for small operators. | [26] | By giving to a central entity the authority of decision making in terms of optimisation and route scheduling for a group of partners who are collaborating, there is no need to increase utilisation of human resources for fleet operators. |
| (F) Significant coordination is needed to achieve data and asset sharing. | [26] | In a centralised structure collaboration scheme, the central coordinator is responsible for coordination of the partners in the collaboration, and the partners are committed to follow central instructions to allow the collaboration scheme to work. |
| (G) Lack of available accurate data. | [16,25,27] | Definition of data structure requirements for collection of unified and accurate data for collaboration. The confidentiality of data collection will be defined through contracts between the partners in the collaboration and the central trustee authority. |
| (H) Lack of trust and common goals. | [11,16] | Use of clear contract agreements, where partners define confidentiality policies, service level agreements, penalties in case of failing, payment conditions, coordination structure, management of unexpected events and contract duration. |
| (I) Lack of a fair allocation mechanism for collaboration revenues. | [11,16,24,28] | Giving different options for revenue sharing to the partners and showing them the cost benefits of each option will allow them to choose, during the negotiation phase, which mechanism will be used for revenue sharing. |
| (J) A neutral third party is required to facilitate collaboration. | [28] | A trustee figure is necessary to implement collaboration. The trustee needs to be a connector between the collaboration partners. Partners might be reluctant to accept a third party, but, this can be overcome through contracts between each partner and the trustee. |

| Barriers/Limitations for Collaboration | Author | Strategies to Overcome Them |
|---|---|---|
| (K) There are clear regional imbalances in freight movement. | [16] | Use the practice of triangulation, where a truck is diverted from its main back route to a third point in order to pick up a return load, potentially increasing the mileage but reducing the amount of empty running. |
| (L) Unawareness of the benefits of participating in collaborative projects. | [29] | Engagement of stakeholders to participate in collaborative networks is crucial. During the initial engagement, it is necessary to show to the possible partners the real benefits of similar collaborative projects. |
| (M) High risk of strategic behaviour in auction collaborative process. | [30] | Effective profit-sharing mechanisms are needed, because these have the potential to impede strategic behaviour. |

Source: [10].

A successful business model should consider into its foundations known, tried and tested enablers for collaboration. These enablers were compiled by [1], identifying opportunities where these enables could be used. Table 2 presents this analysis.

**Table 2.** Enablers and opportunities for collaboration.

| Enabler | Authors | Opportunity |
|---|---|---|
| (A) Common Cultural Mind Set. | [31,32] | The fundamental breakthrough for the success of collaborative projects in the freight industry comes from the willingness of the different industry actors to cooperate. It is critical that partners who decide to collaborate have a common cultural mind-set allowing the implementations of the collaborative process to run smoothly. It is necessary that a fundamental change in the management of transportation sourcing and operations requires that shippers and carriers make an actual "mental shift", de-coupling from their own networks first and then agreeing to re-connect with other shipper network flows. |
| (B) Establishment of Non-disclosure Agreements. | [26,33] | An important way to protect data and assets that are intended to be shared and to assure that owners of the data and assets are willing to provide them to the consortium is to execute non-disclosure or privacy agreements. These may be part of legal contracts or separately negotiated documents. The use of this document will help to increase trust among the partners |
| (C) Stakeholder Engagement. | [26] | It is incumbent upon project leaders and participants in a collaboration project to get to know each other well, establishing a bond and trust between partners prior to collaboration. In this way the partners get to know each other deeply and increase the sense of confidence and trust among them. This will ultimately assure the success of the project. |

**Table 2.** *Cont.*

| Enabler | Authors | Opportunity |
|---|---|---|
| (D) Technology Innovation. | [26] | In many cases, the implementation of a particular technology makes it easier to share data and assets and helps a project to succeed. An automated technology which could accomplish the identification, for instance, of a transportation vehicle without requiring the divulgence of certain data about that vehicle could be a motivator for participants. |
| (E) Articulating Benefits of Sharing. | [26] | It is important for project proponents to be able to explain to the public, to private sector participants, and to other stakeholders how they will benefit from the conduct of the project. Articulating benefits is an important part of project coordination. For instance, publishing analyses of the expected costs savings and benefits of the project reveals openness and transparency such that it could help to assure its success and the involvement of different stakeholders. |
| (F) Legislative Changes. | [23,26] | Normative and jurisprudence aspects of sharing are related to public administrations. Nowadays, the most important facilitators in this category are the different local laws and legislation that help the development of sharing approaches in urban and regional freight transportation. There are two types of approaches: restrictions to non-sharing and incentives to sharing. In the first approach, local authorities could use zero emission zones to force carriers to collaborate with Electric Vehicles (EV) operators to avoid expensive penalties. In the second approach, local authorities could, for instance, incentivise the reduction of empty running through reduced taxation for companies that join collaborative schemes. |
| (G) Previous Relationships Among Partners. | [23] | When participants have already collaborated, because of common interests or because they belong to the same network, transportation sharing is more naturally occurring; it can seem like a step forward in the relationship building among participants. Thus, the trust factor is already in place and the collaboration relationship flows smoothly. |
| (H) Definition of Penalties for Non-Compliance. | [29] | Penalties for non-compliance with contract terms could be made through default payments for each shipment in which a default occurs. Moreover, in some collaborative arrangements, default payments may not be assessed on a shipment basis. The approach used to define the type of penalties for non-compliance with specific terms in a contract will be defined for the collaborative network. The partners who are committed will work with extra care to achieve their liabilities. |

Source: [1].

## 2.2. ICT to Support Sequence-Level Collaboration in Freight Logistics

The concept of horizontal collaboration in freight logistics that has being reviewed in [34] is known for its potential to lead to environmental and economic benefits but can only be achieved with an appropriate technical infrastructure. In this context, it is useful to distinguish between (i) exclusive, focussed collaborations, where two or more operators join forces at a strategic level, and (ii) 'opportunistic' collaboration, where two or more freight operators may not have worked together before but take advantage of temporary alignments in assets and/or delivery locations. For the first type of collaboration, the support infrastructure will develop and specialize over time, becoming highly tuned to the business rules and strategic partnership in question. However, in this article, the focus is on the second, 'opportunistic' type of collaboration, which provides quite different challenges from the

viewpoint of infrastructure but has the potential to impact on the estimated 85% of commercial vehicle miles currently not benefiting from asset sharing [2].

The technical support infrastructure required by opportunistic collaborations can be divided into two clear categories: platform and planning. The platform element is relatively straightforward; it provides facilities for the collaborating freight operators to upload relevant data and to receive outcomes (e.g., collaborative plans) in return. There are several examples of opportunistic freight collaboration initiatives that illustrate the platform aspect (such as Haulage Exchange [35] and Teleroute [36], making use of a wide and familiar range of secure web technologies for data transfer.

In contrast, the planning infrastructure must use specialised planning algorithms at the back end, and the availability and complexity of these algorithms depends on the detailed nature of the collaboration. In the majority of existing 'opportunistic' freight collaboration contexts, the collaboration itself is little more than matching individual loads to individual vehicles and is trivial in terms of the algorithms involved. In the FSLP, however, collaboration is at the sequence level. This type of collaboration, which [34] refer to as 'co-joint routing and which is most allied to 'logistics pooling' in the fine-grained classification offered by [37], has the most potential for significant reductions in cost and emissions, but requires a suitably configured and capable planning algorithm to hand. Research has explored this form of opportunistic collaboration in a variety of contexts; some studies focus on specific sectors (e.g., airport ground operations [38], coastal port operations [39] and grocery retailers [40], while others investigate this in the context of generic road freight [17,28,41]. In all cases, the planning task itself is naturally formulated as a specialized vehicle routing problem [42,43] and is addressed either with mixed integer linear programming [44] or a mathematical formulation based heuristic approach [45] or is addressed with a concoction of metaheuristics [46]. Additionally, a recent study [47] provides empirical evidence related to transportation planning and distribution network design used in conjunction with commodity considerations, which might be a source of sustainable supply chain performance.

It is notable, however, that the solution methods reported in the literature focus on the algorithms required for planning the joint operations of a pre-selected pair (or wider group) of operators. The higher-level, a-priori task of partitioning a larger collection of potential collaborators into smaller collaborating groups is rarely addressed. The approach used in this paper is outlined in Sections 4.1 and 4.2. Meanwhile, for detailed planning, a metaheuristics approach is being used, and is also outlined in Section 4.2.

## 3. Gain-Sharing Business Model

The literature review presented in the previous section evidences a gap between the organisational theory of collaboration and the need of ICTs that understand the foundations of collaborative networks at their core, to enable operational and functional tools that truly support collaboration processes and its win-win philosophy. With the aim to start bridging this research gap, Section 3 presents details of the gain-sharing business model focusing on the organisational aspects of the proposed collaborative business model, Section 4 presents details of the proposed algorithms needed to be implemented on the functional tool, and Section 5 presents the bridge between the collaborative business model and the algorithms that support this business model and are the back-end of the FSLP.

The business model introduced in this section clarifies that there is a win-win for fleet operators as well as their customers, where "coopetition" can be delivered through a collaboration platform that yields significant commercial benefits for all participants [1]. An animation with the proposed business model is provided on Supplementary Materials.

### 3.1. Definition of the Collaboration Process Using a Combined Coordination Structure

The proposed collaboration process defined in [10] uses a combination of the different coordination structures (discussed in Section 2.1.1), as outlined in Figure 1. It is proposed that the centralized coordination is managed and harmonized by a neutral trustee, while the decentralised coordination

is achieved by the different members of the CN in specific activities. This approach will bring the benefits of the different coordination structures to the different phases of the collaboration process and therefore bring flexibility to enable its implementation between different parties. The details of this process are provided in Section 3.3. It is also important to clarify that, at this stage, the proposed business model focuses on the horizontal collaboration between possible competitor freight operators. Future developments of the proposed business model might aim to achieve multilateral collaboration involving shippers as well.

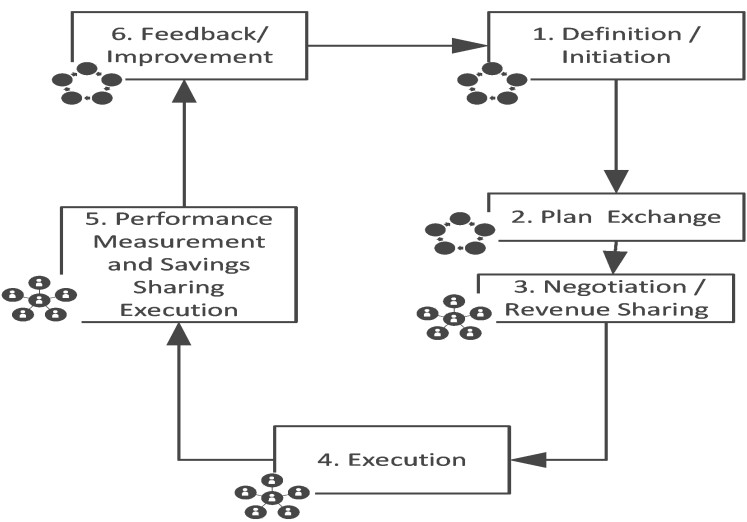

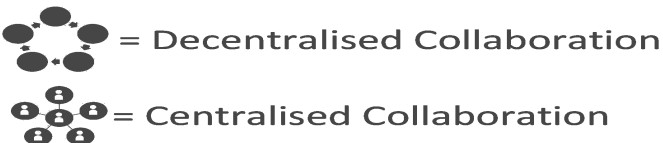

**Figure 1.** Definition of collaboration process using a mix coordination structure.

## 3.2. Overcoming Barriers and Boosting Enablers for Collaboration

The proposed business model has been derived following an analysis of the main barriers and limitations found in the literature and the strategies used to conquer them (Section 2.1.2—Table 1). Based on these foundations, the FreightShare Lab (FSL) business model will be:

1. Implementing the use of a neutral trustee (28) to facilitate the collaboration process between competitors by managing and coordinating day to day operations, and to ensure the confidentiality of the members' structured data [27] and compliance with competition law [16,23–26]. The barriers covered by these implementations are: (D), (E), (F), (G) and (J).

2. Defining contractual agreements where partners negotiate to establish confidentiality policies, service level agreements, performance failure penalties, payment conditions, the coordination structure, the management of unexpected events and contract duration [11,16,24,28], fair mechanisms for the allocation of collaboration revenues [11,16,24,28], building trust and the setting of common goals [11,16]. The barriers covered by these implementations are: (B), (C), (H), (I) and (M).

3. Raising awareness of the potential benefits of being part of the collaborative ecosystem (29) by creating, for instance, a virtualisation tool that allows interested parties to engage from the beginning. The barriers covered by these implementations are: (A) and (L).

The FSL project drew on the literature related to enablers for collaboration to ensure a successful initial engagement of stakeholders (Section 2.1.2—Table 2) by:

1.  Issuing personal invitations to attend workshops and clinics to validate the virtualisation tool and provide feedback related to the proposed processes, especially to those stakeholders that the consortium has had previous experiences of collaboration [23,26]. The enablers covered by these implementations are: (C), (D), (E) and (G).
2.  Selecting those stakeholders with an open mind towards collaboration [31,32]. The enabler covered by this implementation is: (A). Ensuring the confidentiality of stakeholders' data by establishing Non-Disclosure Agreements (NDAs) to increase trust among them [26,33]. The enabler covered by this implementation is: (B).

## 3.3. Definition of the Gain-Sharing Business Model

The traditional business model canvas developed by [48] is constituted by nine building blocks that are specifically designed for organisations who operate individually. However, the proposal for the FSL is to adapt the traditional business model canvas through additional building blocks so as to reflect the collaborative approach to business operations. The concepts are defined in Section 2.1.1; forms of collaboration and strategies for collaboration have been included to detail how this is managed and fulfilled from an operational perspective. Figure 2 presents the adapted business model canvas for the FSLP. Comparison of the proposed model with other implemented business models in the context of collaboration is recommended as a next area of research to understand the viability and benefits of the current proposal.

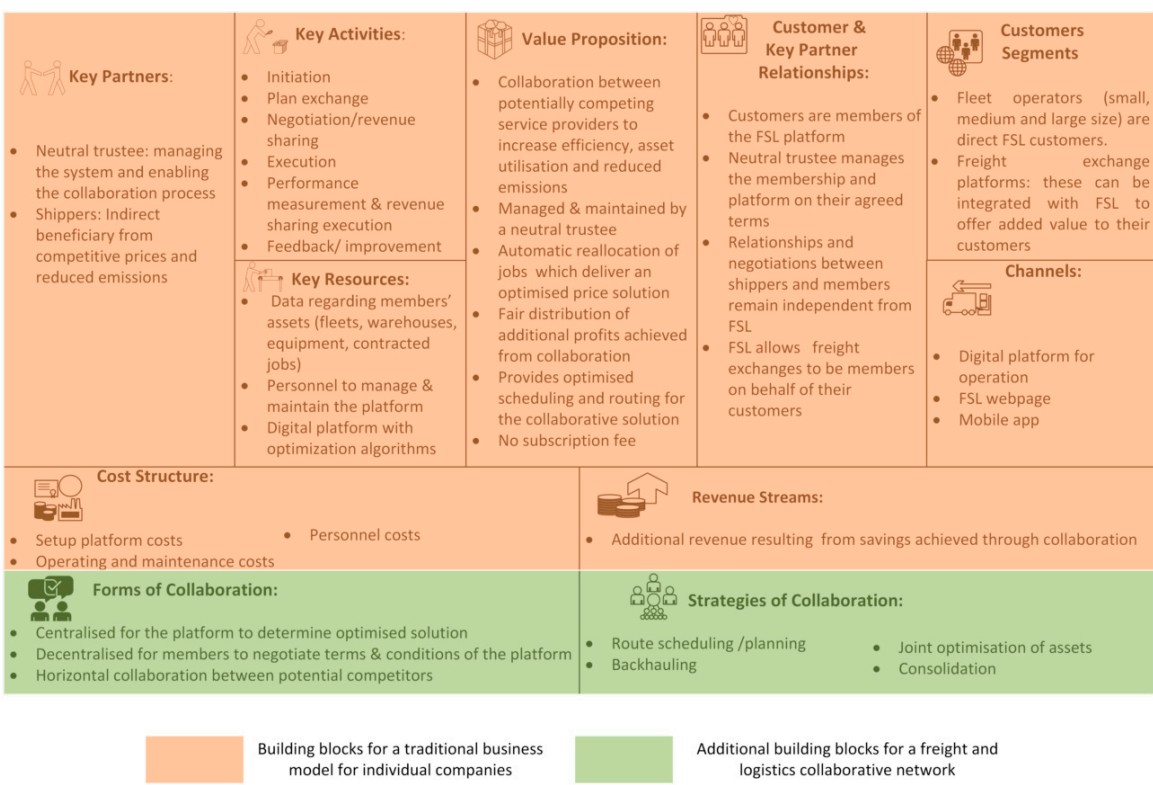

**Figure 2.** Adapted business model canvas for the FreightShare Lab Platform (FSLP) from [10].

The eleven building blocks proposed above for the FSLP are described in detail in [1]. Table 3 provides a consolidation of the basic insight needed to understand how the different building blocks constitute a key element for the proposed business model.

**Table 3.** Description of building blocks defined for FSLP.

| Building Block | Description |
| --- | --- |
| Value proposition | The objective of the FSL platform is to increase competitiveness, efficiency and utilisation, by creating a collaborative ecosystem. The platform will search for the most efficient delivery in terms of operating costs, fleet utilisation and emissions for the fulfilment of contracts submitted by members of FSL using the assets of FSL members. A wide geographic coverage of members' assets increases the likelihood of capacity being available and in turn increases the chances of fulfilling more contracts in any given period of time. However, the FSL algorithm will only reallocate jobs where price and emissions are lower than those possible if performed by the Contract Holder (CH). In the event that a lower price and lower emissions cannot be found, the CH would then fulfil the contracted job. |
| Customer Segment | Fleet operators are FreightShare Lab's principal direct customers. Given their competitive nature, and to ensure compliance with competition law, it is necessary for a neutral trustee to facilitate the collaboration between them. Where the fleet operators or their customers participate in specific load-sharing, auction, return-load freight exchange platforms, FSL will look to provide value added for them as well: FSL will offer a collaborative relationship by which those contracts awarded through these other platforms can be uploaded into FSL system to see if a better alternative can be found and arranged (i.e., at a lower price and reduced emissions); in effect, the FSLP acts as a 'platform of platforms'. |
| Customer Relationship | A neutral trustee is an organisation responsible for ensuring the collaborative network will be constructed in such a way that a fruitful long-term, sustainable relationship is established and maintained. Partners in a collaboration agreement (possibly competitors) could provide commercially sensitive data to the trustee organisation, which can maintain the required confidentiality and security of such data use, according to contractual terms and conditions agreed with the data owners, for fulfilling the purposes of FSL. In this way, compliance with EU competition and data protection laws is provided. Arguably, the platform is best managed as a cooperative by the FSL members; all terms and conditions, rules of the FSL business, quality and standards shall be agreed in a decentralised manner by the members and profits distributed among them. FSL members would upload contracts they have individually agreed with their customers into the platform, securely and confidentially, for the system to analyse. |
| Channels | A digital platform will be used to support and facilitate the operation of the FSL. Regular meetings will be necessary between partners and trustees to build trust, incentivise collaboration and negotiate terms and conditions. |
| Key Activities | The definition of the key activities in a collaborative process has been proposed in [10,49] based on previous ideas from [5,19,24,50–53]. These activities are: (1) Initiation, (2) Plan Exchange, (3) Negotiation/Revenue Sharing, (4) Execution, (5) Performance Measurement and Revenue Sharing Execution and (6) Feedback/ Improvement. |
| Key Resources | Data and assets provided daily by the FSL members in accordance with the terms and conditions negotiated for each cluster. The digital platform with the back-end optimization algorithms and the support team to manage and maintain the platform and to allow the orchestration of collaboration among the members |

**Table 3.** *Cont.*

| Building Block | Description |
|---|---|
| Key Partners | In this model, shippers, do not have direct access to the platform, but will benefit from it through sustainable competition among the logistics service providers and lower emissions associated with the fulfilment of their jobs. The relationships, interactions and negotiations between shippers and carriers remain the same. |
| Revenue Streams | The FSL algorithm will reallocate jobs where the total operational cost and associated profit margin is lower than the operational cost of fulfilling their own contracted jobs with their own logistics assets. This will guarantee that all collaborating members, both the original CH and Contract Performer (CP), maintain their agreed profit margins, as well as providing them with additional revenue. |
| Cost Structure | The business model provides that all savings achieved through the platform will be shared between members, once the costs of running the platform are covered. Costs will be distributed across collaborating members in proportion to the savings they generate, and hence those members that do not participate in any transaction will neither incur any costs during that period nor share any additional revenue generated. |
| Forms of Collaboration | A combination of decentralised and centralised coordination in each key activity is proposed in order to bring wider benefits and flexibility to enable its implementation between different members. The centralised coordination allows the platform to determine the optimal solution and assign to specific operators. The decentralised coordination facilitates members to negotiate terms and conditions for each cluster. |
| Strategies of Collaboration | The strategies imbedded into the optimization algorithms in the back-end of the platform are: route scheduling/planning, join optimization of assets, backhauling and consolidation. The use of complementary strategies into the platform makes the FSL more attractive for operators, enabling them to find the most cost and environmentally efficient solutions. |

In addition to the description of the business model canvas proposed for the FSLP, it is worth explaining the basis for the gain-sharing business model in terms of operations, economics and benefits. The FSLP tries to find the optimum logistics solution that can fulfil contracts at a lower price and with lower emissions by exploiting the combination of assets that is available to the FSLP. The proposed operational and economic view of the business model ensures that both the original Contract Holder (CH), and those companies that provide the assets deployed by the FSLP to fulfil the contract (Contract Performer (CP)) cover operating costs, retain their profit margins and receive a bonus from the difference between the estimated operating costs of the CH and CP (explained in Figure 3).

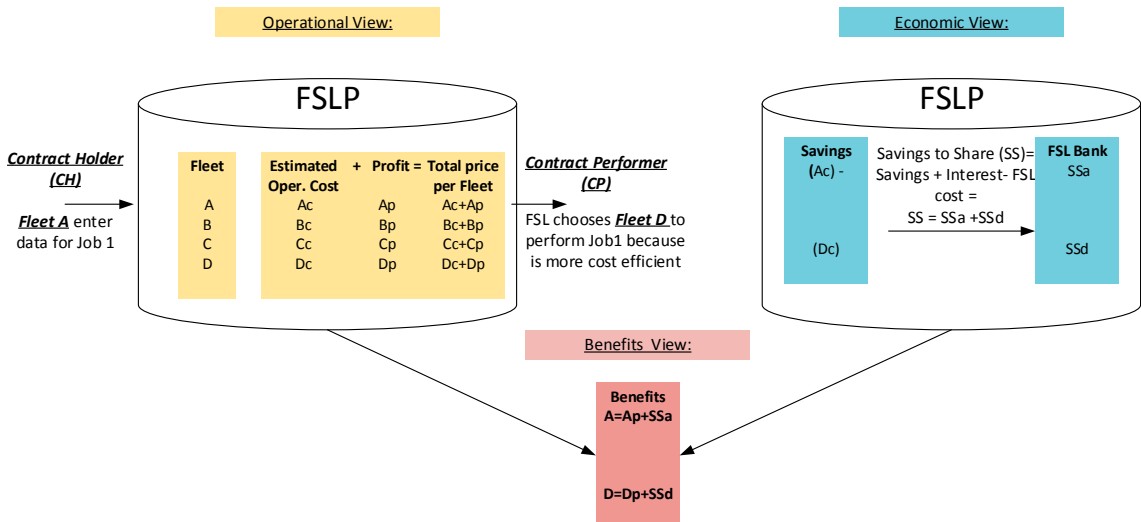

**Figure 3.** Schematic representation of the operational, economic and benefits views of the FSLP adapted from [10].

## 4. Special Planning Algorithms for Logistics Collaboration

Many fleet operators in the freight industry perform operations in a daily cycle; for instance, operations are planned on day one, typically in the evening, and executed on day two. Delivery plans for Thursday (for example) will typically be calculated on Wednesday, based on up-to-date customer orders and availabilities, and then be sent to drivers in advance. The FSLP aligns with this general practice; every day, data relevant to tomorrow's plan is collected, up to an agreed cut-off time (e.g., 6 p.m.) [1]. Therefore, the routing is planned differently each day depending on demand and incorporating any time window restrictions for drivers [49,54]. Then, collaborative delivery plans are derived and sent to the participants for distribution to drivers and warehouses in advance of their execution the next day. Further iterations of the FSL could develop platforms which handle night work independently, or multiday planning horizons, each focusing on a distinct subset of members [1].

FSL members who use the FSLP can upload either, or both, of two datasets:

1. **Vehicles** that the member wishes to make available to the platform
2. **Jobs** that the member wishes to have processed by the platform.

The member also indicates the type of sharing arrangements that are appropriate for them. These daily data uploads, as well as the collection of solutions, may be done either manually—via the manipulation of csv files or via an Application Programming Interface (API)—facilitating the interaction between the FSLP and participants' IT systems—depending on the resources available. To fully automate the process, the API essentially enables members to send their jobs, vehicles and sharing data to the FSLP as a JavaScript Object Notation (JSON) packet, and to check whether a solution is available (and, if so, receive that solution in a format that the member can pre-configure).

### 4.1. Share-Cluster Assignment Algorithm

When new FSL participants register with the FSLP, they are assigned to a 'share-cluster'. A share-cluster, in this context, is a collection of FSL members that are likely to benefit from asset-sharing based on geographical proximity and the compatibility of the freight categories they handle. An 'affinity score' for each existing cluster is calculated, and a new participant joins the cluster with the highest affinity score. The idea of the affinity score is to provide a fast estimate of the potential for two freight operators to work together fruitfully. The score is 0, for example, if neither of the operators' vehicles are able to carry the goods handled by the other operator, or if their depots are simply too far apart for collaboration to be sensible. Meanwhile, the score will be at or near its maximum [1] for

freight operators whose depots are very close (within a few minutes of travel), and who carry similar categories of freight, and whose customers tend to be aligned geographically. Hence, the potential benefits of two fleets working together is partly dependent on the geographical disposition of those fleets' customers; however, no data on this will be available to the FSLP when a participant initially registers, so their customer base is initially estimated to be at postcodes randomly scattered within 100 km of their depot. As time progresses and customer data for a new participant becomes available, that participant's affinity scores for each existing share-cluster will be recalculated, and the share-cluster assignment may change as a result. The detailed steps of share-cluster assignment are provided in Figure 4.

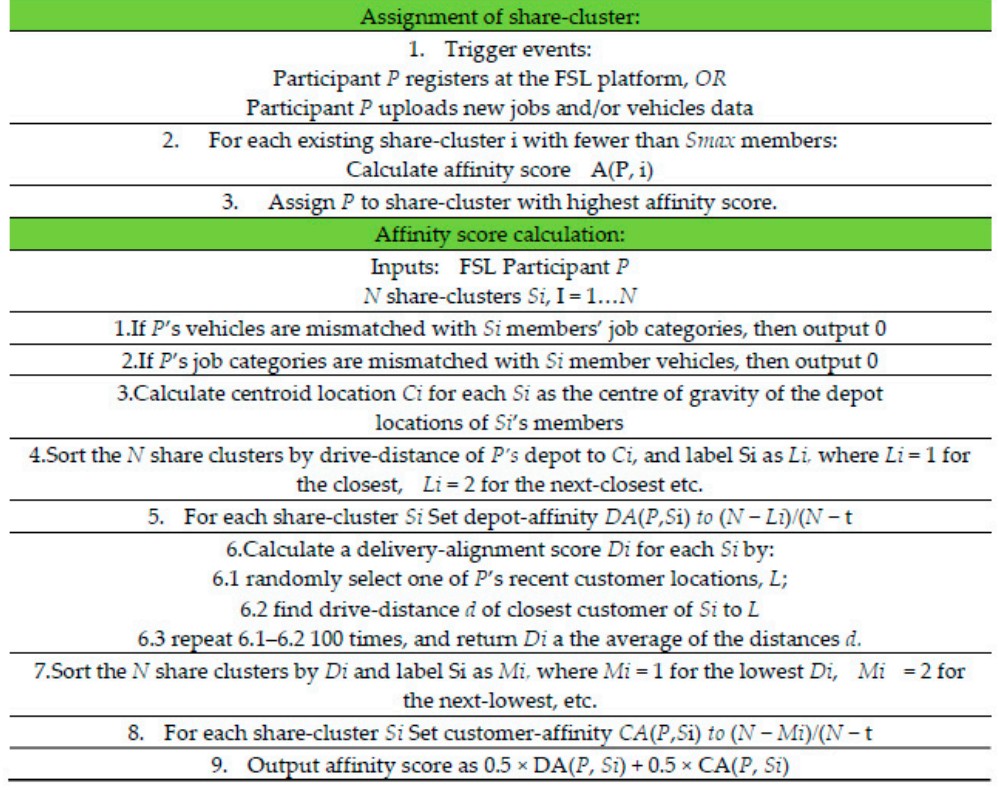

**Figure 4.** Pseudocode for the algorithms and routines that assign an FSL participant to a share-cluster when that participant registers and adapts that assignment when new data is uploaded.

### 4.2. Collaborative Optimization Algorithm

This algorithm is explained in [1], with additional details also provided here in Figure 5. At the cut-off time on any given day, the FSLP will have a dataset D = (V, J, S), respectively denoting the full set of vehicles and jobs available, and the corresponding sharing arrangements. It may be tempting to view D as a single large-scale vehicle routing problem with specialised constraints [42,43]. However, the typical scale of D (e.g., 20,000 jobs, 5000 vehicles) compromises the ability of current algorithms to address this in a reasonable timeframe. Therefore, the FSLP instead operates a 'divide-and-conquer' strategy to partition D into a series of smaller problems, D1, D2, ... , Dn, and then solves each of these problems in turn.

| Collaborative Optimization |
|---|
| 1. Trigger event: <br> Time *T* has been reached, where *T* is deadline for FSL participants to upload <br> their final data for planning |
| 2. For each share-cluster *Si*, run *FindBestPartition (Si)*, which returns a set of fleet-groups {*G1,* <br> *G2, ... Gn*}, each containing either 1, 2 or 3 fleets; |
| 3. For each fleet group, run a suitably modified commercial fleet-planning algorithm to obtain an <br> optimized collaborative plan for that group. |
| Find Best Partition |
| Inputs:   A share-cluster, Si, with *N* member fleets, F1,...FN,   and a time-limit *TL* |
| 1. Generate a random initial partition *PT* = {*G1..Gn*} and calculate its *fitness f(PT)* as the mean <br> share-factor across its groupings; |
| 2. Repeat the following until time-limit *TL* is reached: |
| 2.1   Perturb the partition *PT* to obtain a new valid candidate partition *MT;* |
| 2.2.   Calculate *f(MT)* as the mean *sharefactor* of its groupings; |
| 2.3.   If   *f*(MT) is better than *f*(PT), replace *PT* with *MT* |
| 3.Return *PT* = {*G1,...,Gn*} as the best partition |
| Calculate share factor |
| Inputs: A grouping of FSL participants *G* = {*P1,...,Pn*} |
| 1. For each job *j* submitted to the portal today by each participant *Pi* in *G,* <br> calculate sharescore ss(*G,j*) |
| 2. Initialise sharefactor(*G*) as the proportion of ss(*G,j*) values that are less than 1 |
| 3. Add penalty value −1×*n* if sharefactor(*G*) is 0. |
| Calculate share score |
| Inputs:   A group *G of n* FSL participants, and a job *j* uploaded by FSL Participant *Pi* in that group |
| 1. Calculate *Ts*, the time it would take to service *j* from *Pi*'s depot |
| 2. For each participant *Pj* in *G*, where *i* is not equal to *j* |
| 2.1. Calculate *Tc*, the time it would take to service *j* from *Pj*'s depot |
| 2.2. Record the value *Tc/Ts* |
| 3. Output the proportion of *Tc/Ts* values calculated in Step 2 that are < 1. |

**Figure 5.** Pseudocode for the Collaborative Optimization algorithm.

The starting point for the partitioning into smaller problems is the current set of share-clusters (discussed in Section 4.1), which may typically contain 10–20 fleets. To identify the ideal groupings for this particular day, the partitioning strategy makes use of a specialised metric called the sharefactor, which predicts the extent to which two FSL members would benefit by working together. Although highly simplified, asset sharing is effective to the extent that fleet A's orders are geographically more convenient for fleet B's vehicles to handle. This notion is estimated for each job by a sharescore. By example, fleets A and B submit their data (VA, JA, SA) and (VB, JB, SB) to the FSLP: Their respective sharing arrangements are compatible (e.g., A's vehicles can carry B's jobs and vice versa). The sharescore ss(j) for each job from JA is defined as follows: ss(j) is Tc/Ts, where Tc is the time it would take for a vehicle from VB to process the job if it were located for pickup at B's depot, and Ts is the time it would take a vehicle from VA to process it from A's depot. A sharescore below 1 suggests a time and mileage advantage. Moreover, the sharefactor (A,B) for two fleets is the proportion of the combined jobs (from both JA and JB) that have a sharescore below 1. The larger the sharefactor, the larger the potential benefits for collaboration, suggesting that an initial shuttling of orders between the depots could result in more efficient delivery and more than compensating for the shuttling costs.

Using the sharefactor calculated for all groups of fleets, a fast filtering algorithm ranks the potential groupings of fleets in terms of the potential resource savings that can be achieved from collaboration. The FSLP then considers each of these groups in turn and resolves the associated fleet planning problem arising from combining their assets, according to the declared sharing arrangements among the group.

The fleet planning solver used by the FSL is a variant of commercial software that is currently operating in the (single) fleet planning industry. Consistent with state-of-the-art algorithms of its

type, its design combines various aspects of metaheuristics search [46], many-objective search [55] and traditional AI planning [56]. The range of factors considered by the algorithm are those with material impact on time, costs and mileage, including:

1.  Costs: Cost-per-mile, potentially different for different vehicles; driver cost-per-hour (including any overtime); a fixed cost per vehicle; an 'Opportunity Cost' of not delivering a job, commonly supplied by users of fleet optimisation software which can be considered a penalty fee for a fulfilment failure.
2.  Times and associated constraints: driver shift times and working time constraints; time windows for pickup and delivery of each job; service times for pickup and delivery of each job; driver briefing time; realistic times for every journey and vehicle type.
3.  Capacity issues: weight and volume capacity of each vehicle, weight and volume of each job, ensuring vehicles are never overloaded.

The solver produces a detailed schedule of activities, specifying an itinerary for all or some of the vehicles involved, much like the itineraries typically delivered by fleet optimisation software. The schedule for a group of fleets (typically two or three) will usually process all the jobs involved in the group, although this is not always possible. However, in such circumstances, a collaborative schedule will always be able to process at least as many jobs as would be achievable without asset sharing.

### 4.3. Sharing Scenarios

When the fleet planning solver works with a specific group of fleets, the first step is to identify which style of collaboration will be used. This will either be 'no special arrangements' (NSA), 'morning transfer' (MT) or 'consolidation centre' (CC). In some cases, this decision is determined directly by the registered preferences of the fleets in question (e.g., they can only use the CC approach if this specific collection of fleets has an agreed collaboration centre site registered on the FSLP). In other cases, two or more of these modes may be applicable. If NSA is one of the two or more styles, it is simply discounted, because our background work has identified it to be almost always inferior to MT and CC. Meanwhile, if both MT and CC are valid candidates for this group of fleets, a fast heuristic assessment is made, whereby a smaller version of the joint planning problem is formulated and solved using each in turn, and the style achieving the best results will be used for the full-scale planning.

In the NSA scenario, every job will be picked up from its starting location (usually a depot) and delivered directly to its destination (usually a customer site), just as in a single-fleet solution; however, while the depot and customer will be associated with one FSL participant, the vehicle actually picking up and delivering the job may be that of another FSL participant. In the NSA scenario, collaborating fleets will have established the minimal sharing arrangements, which allow each other's vehicles access to each other's depots.

The CC scenario represents relatively mature arrangements between a pair of fleets, whereby a consolidation centre location has been agreed for exchanging their goods. Typically located at a central point between their depots, each fleet will move some of their jobs directly to this site to be picked up and taken to their destination by the other fleet.

From the data analysed thus far, the MT scenario is the most beneficial arrangement, representing more efficient outcomes (compared with NSA) with relatively low business complexity (compared with CC); in MT, fleet A first moves some of their jobs (those most convenient for fleet B to deliver) to fleet B's depot; fleet B proceeds to deliver those jobs along with its own jobs, and vice versa.

The fleet planning solver first determines which sharing method to use (based on fleets' preferences and/or a preliminary fast optimization using a reduced version of the problem) and then chooses the method. In the case of CC or MT, the solver will calculate the appropriate subset of fleet A's jobs for fleet B to carry, and vice versa. Vehicles from each fleet then transfer these jobs from their pick-up locations to the consolidation site, or to the other fleet's depot, depending on the scenario. Taking

the updated locations of assets and time and mileage as the starting point, the planning process then proceeds as if the two fleets were a single operator.

### 4.4. Validation

Proof-of-concept studies, reported in [2], using synthetic data, showed that asset-sharing could achieve cost and emission savings ranging from 16% to 53%, with a mean of 19% when just two fleets work together and more emphatic savings in denser road networks (e.g., European vs. US). Additionally, up to five fleets working together achieved savings of 70% with diminishing returns thereafter.

To further validate the theory and comprehend the extent to which collaboration, and in particular the proposed gain-sharing business model, would be beneficial both for the economy and fleet operators, the FSL project tested the algorithms with real-world fleet data and incorporated the gain-sharing business model described herein. To this end, real vehicle and jobs data was harnessed from 125 UK fleets (with the aid of Route Monkey Ltd. (Glasgow, Scotland) and Trakm8 PLC (Birmingham, West Midlands, UK)), several of which provided several days' worth of data. With these data, a simulated single day data submission to the FSLP was constructed, each involving genuine location, vehicle, freight-category, capacity and volume data. The largest simulated single-day FSLP submission that could be constructed from the available data included 116 fleets. The fleets varied in size from 5 to 300 vehicles, and the number of jobs per fleet varied from 22 to 800.

### 4.4.1. Efficiencies Achieved

Baseline results were produced for each individual fleet in a standard, non-sharing scenario. Then, running the grouping and filtering algorithms on this dataset led to 30 pairs of fleets which were predicted to benefit from sharing. The modified solver then ran on each of those pairs to identify a precise sharing itinerary in each case and quantify the benefits in comparison with the single-fleet scenario. Significant benefits were obtained for 25 of the 30 pairs. In practice, the FSLP will also provide the participant with its single-fleet solutions for cases where sharing is not beneficial on that specific day.

The 25 fleet pairs that benefited from sharing comprised 6700 jobs. The detailed sharing outcomes indicated significant commercial benefits from the FSL collaborative business model, combined with important increases in profits, as shown in Figure 6. Outcomes were shown in [1], the total daily profits aggregated for all the fleets collaborating in FSL, both for those which FSL identifies an optimised solution through collaboration, and for those for which it does not, under each of the three collaboration scenarios. The results indicated that a high proportion of jobs can be delivered at a lower cost than that of fleets operating independently.

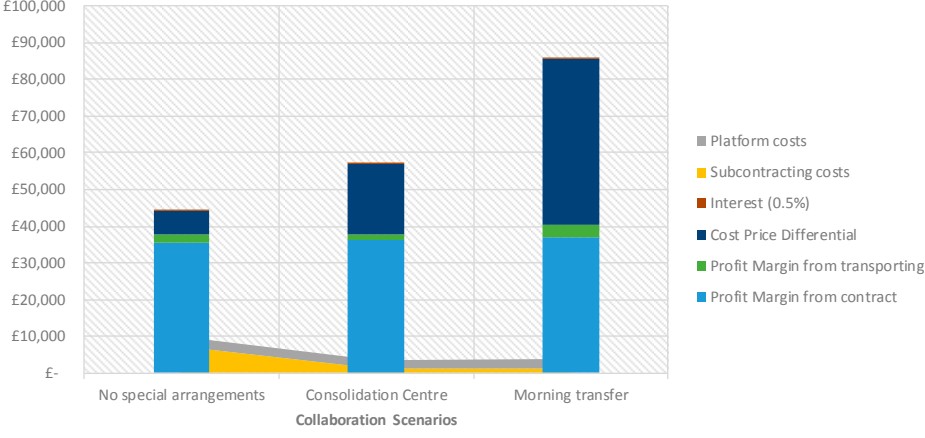

**Figure 6.** Total daily Revenue Streams with FSL [1].

Furthermore, the collaboration made further capacity available with a corresponding increase in the number of jobs that could be delivered in a single day. Table 4 presents the results found in [1] for the three collaboration arrangement scenarios.

**Table 4.** Efficiencies achieved through the different collaboration arrangements.

|  | No Special Arrangements | Consolidation Centre | Morning Transfer |
|---|---|---|---|
| **% of jobs with savings** | 48% | 63% | 72% |
| **% increase of jobs completed per day** | 3% | 1% | 5% |
| **Daily mileage reduction** | −3012 km | 3523 km | 2143 km |

Source: [1].

It is also illuminating to consider the benefits in fleet utilisation that arise from sharing; however, this may be misleading, subject to circumstances on a case by case basis. Defining 'utilisation' simply as the number of jobs processed per vehicle (which acts as a broad proxy for capacity utilized per vehicle), results show that utilisation was improved by 1.5% across the 25 fleet pairs above, with 12 pairs gaining improved utilisation, and 6 pairs having an overall lower utilisation. However, the cases of reduced utilisation were invariably cases in which the sharing scenario led to significantly more jobs processed; in such cases, utilisation fails to be a useful indicator because the difficulty of reaching the latter jobs does not influence the no-sharing utilisation figure. Considering only cases in which there was no difference in the number of jobs processed, the results show a mean of a 3.5% improvement in utilisation with sharing, ranging from 0 to 9.5%.

### 4.4.2. Wider Socio-Economic and Environmental Benefits

In addition to the private costs generated by fleet operators, their activity imposes externalities on society and the environment. Optimised truck journeys through collaboration will lead to a reduced total distance travelled and a reduced number of trucks on the road with a consequent reduction in environmental and social costs.

The initial algorithm results presented in [1], and shown in Tables 5 and 6, have been utilised to quantify the annual change in these external costs and hence, if reduced, offer an understanding of the level of benefits that can be expected through collaboration. The changes were calculated based on reduced mileage, following the UK DfT's WebTAG unit A5-4 (DfT, 2018) and TRL publications for the reduction in emissions (TRL, 2009).

**Table 5.** Annual mileage and emissions savings through the different collaboration scenarios.

| Collaboration Arrangement | Emissions Saved | | | |
|---|---|---|---|---|
|  | $CO_2$ (tonnes) | PM (Kgs) | NOx (Kgs) | HC (Kgs) |
| **No Special Arrangements** | −629 | −122 | −12 | −996 |
| **Consolidation Centre** | 735 | 143 | 14 | 1165 |
| **Morning Transfer** | 447 | 87 | 9 | 708 |

Source: [1].

**Table 6.** Annual wider economic costs savings through the different collaboration scenarios.

| Collaboration Arrangement | Other Wider Economic Cost Savings | | |
|---|---|---|---|
| | Congestion (k£) | Infrastructure (k£) | Accidents (k£) |
| **No Special Arrangements** | −211 | −107 | −3 |
| **Consolidation Centre** | 247 | 125 | 4 |
| **Morning Transfer** | 150 | 76 | 2 |

Source: [1].

The 'Morning Transfer' scenario leads to the highest cost efficiencies and therefore profits, however, mileage reduction is less than that achieved from the consolidation centre scenario, which leads to wider economic cost savings. The 'No Special Arrangements' scenario leads to additional mileage due to the journeys vehicles have to do to get to other fleets' depots, thus generating additional emissions and wider economic costs. Based on these results, the use of NSA has been ruled out in the FSLP, unless it is the only mode available.

## 5. Integration of the Gain-Sharing Business Model with the FSLP Algorithms

Founded on the business model described in Section 3 and the definition of the two algorithms in Section 4, Figure 7 presents a simple workflow, integrating both within the FSLP.

A detailed description of the different stages of the collaboration process is presented in Table 7.

**Table 7.** Description of building blocks defined for FSLP.

| Collaboration Process Stage | Description |
|---|---|
| 1. Collaboration Initiation | The collaboration process starts when a logistics operator submits an 'Expression of Interest' (EoI) to the platform. The platform, which is managed by the neutral trustee (represented on the workflows), receives the EoI and performs the "Share-Cluster Assignment Algorithm" to identify a suitable 'cluster' of members in terms of equipment, geographical location, type of products, compatibility and data transfer capability, among others. If such a cluster exists, the interested party will receive information relating to the default contract pertaining to that cluster, to which it has been assigned, as well as its legal and commercial terms and rules of collaboration. Where a new member enters an existing cluster, they will not be able to negotiate or alter the previously defined terms and conditions of collaboration within the cluster, and will therefore have to accept any existing conditions until the following Ordinary Meeting, where they will have full voting rights and the opportunity to re-negotiate. Following the acceptance of conditions, the interested party will sign a contract with the cluster and another contract with the neutral trustee, where contracts are sent and signed electronically. In this way, the party becomes a new FSL member. If no suitable cluster exists for the interested party to collaborate with, the platform will check whether there is a member "on hold" and waiting for a possible partner that matches the appropriate criteria. If this match is affirmative, the two matched parties will perform the following two-step process; Plan Exchange and Negotiation. If there is no match found, then membership will be put on hold until another suitable collaboration partner eventually shows interest in FSL. |

**Table 7.** *Cont.*

| Collaboration Process Stage | Description |
| --- | --- |
| 2. Plan Exchange | At this stage in the collaboration process, the parties that were matched through the platform need to agree with the Plan Exchange; the sharing of data on the vehicles and jobs that are available through the platform using an Application Programming Interface (API) and the practicalities for execution of the API and the time limit. The parties are also informed about the operating costs, as calculated by the platform, based on standard tables. They can then decide if they want to use those standard operating costs or if they want to provide their own and real operating costs. This last option is advisable for those operators that are very efficient in their operations and are willing to embrace as many jobs as possible. Lastly, the parties are informed about their voting rights, frequency of vote and the next planned meeting where they will have the opportunity to re-negotiate any relevant terms and conditions. |
| 3. Negotiation/Revenue Sharing: | At this stage, the collaborating parties that were matched through the platform will negotiate terms such as: Service Level Agreements (SLA), liabilities, level of profit margin, rules of collaboration and the revenue sharing mechanism. This stage will not be applicable to those parties that have already been matched with an established cluster where, instead of negotiating terms and conditions, the parties shall accept those already in place but have the possibility of re-negotiation at a later stage. Any re-negotiation will take place at the Ordinary Meeting at the end of each financial period, where members will execute their voting rights to determine any terms ahead of the next Renegotiation of Conditions. If a new cluster is formed, the platform will provide a default contract with the terms that need to be agreed between the parties, thus providing them with a template for negotiation. If an agreement is reached among them, they will sign two contracts, one with the cluster and one with the neutral trustee. In this way, the parties officially become part of the FSL consortium. If an agreement is not reached in the negotiation phase, the process will terminate for both of them or otherwise will be put on hold to wait for another possible match |

**Table 7.** *Cont.*

| Collaboration Process Stage | Description |
|---|---|
| 4. Execution; | The execution will be led by the neutral trustee who will be responsible for managing and maintaining the platform. The FSL members will submit data through the API, in terms of the available assets and the contracted jobs they wish to post, which will only be visible to the neutral trustee, thus ensuring commercial confidentiality between members. This data will have to be entered into the API before a pre-determined time (e.g., 6 p.m. the day before a particular delivery contract is to be fulfilled). The platform will then run the algorithms and send the operators the resulting collaborative schedule and route. Once the schedule is received by the members' transport planners, they can accept or reject the master schedule. If any of the members reject one or more of the jobs, the platform will adjust the solutions without the jobs rejected and recalculate schedules, routes and prices. An upper limit of 2% of the total offers to fulfil a contract (in whole or in part) may be rejected by each member in order to avoid too many re-calculations that may delay the schedules being received in sufficient time to organise the required resources to fulfil the job request. The updated solution is to be final; the members are bound to accept it and act accordingly (as would be specified under the terms and conditions they have agreed to when joining the FSL). On the day that deliveries are due, drivers will receive the schedule and route via the driver app. The drivers will then fulfil the jobs and send a Proof of Delivery (PoD) through the app. If everything goes as expected, the platform will send an electronic invoice to the CH corresponding to the jobs posted into the platform and performed by the CP. This e-invoice will reflect the standard operating costs that, according to the algorithm, would have been incurred by the CH if it were operating without collaboration. The platform will also make the corresponding payment and send an electronic deposit slip to the CP that fulfils the job. The latter payment will be equal to their operating costs, as estimated by the algorithm, plus their previously agreed profit margin. The platform retains the difference between the amount paid by the CH and the corresponding amount paid to the CP. |
| 5. Performance Measurement and Revenue Sharing Execution | The cost differential, being the difference between the CH payments to the FSL platform and the corresponding payment by the FSL platform to the CP, is retained until the end of the financial year. At the end of the financial year, the neutral trustee executes the gain-sharing for each cluster after deducting the platform management fees and running costs incurred throughout the financial year. The gain-share is made directly to members' bank accounts at the end of the financial year in accordance with the cluster rules. |
| 6. Feedback/Improvement | The aggregated results in terms of participation, market share, operational and financial performance for the financial year, are presented at the Ordinary Meeting, where members may re-negotiate the terms within each cluster. Those members who agree with the changes to the terms and conditions may continue to be members of FSL and to work collaboratively, while those who do not accept them will leave the cluster. |

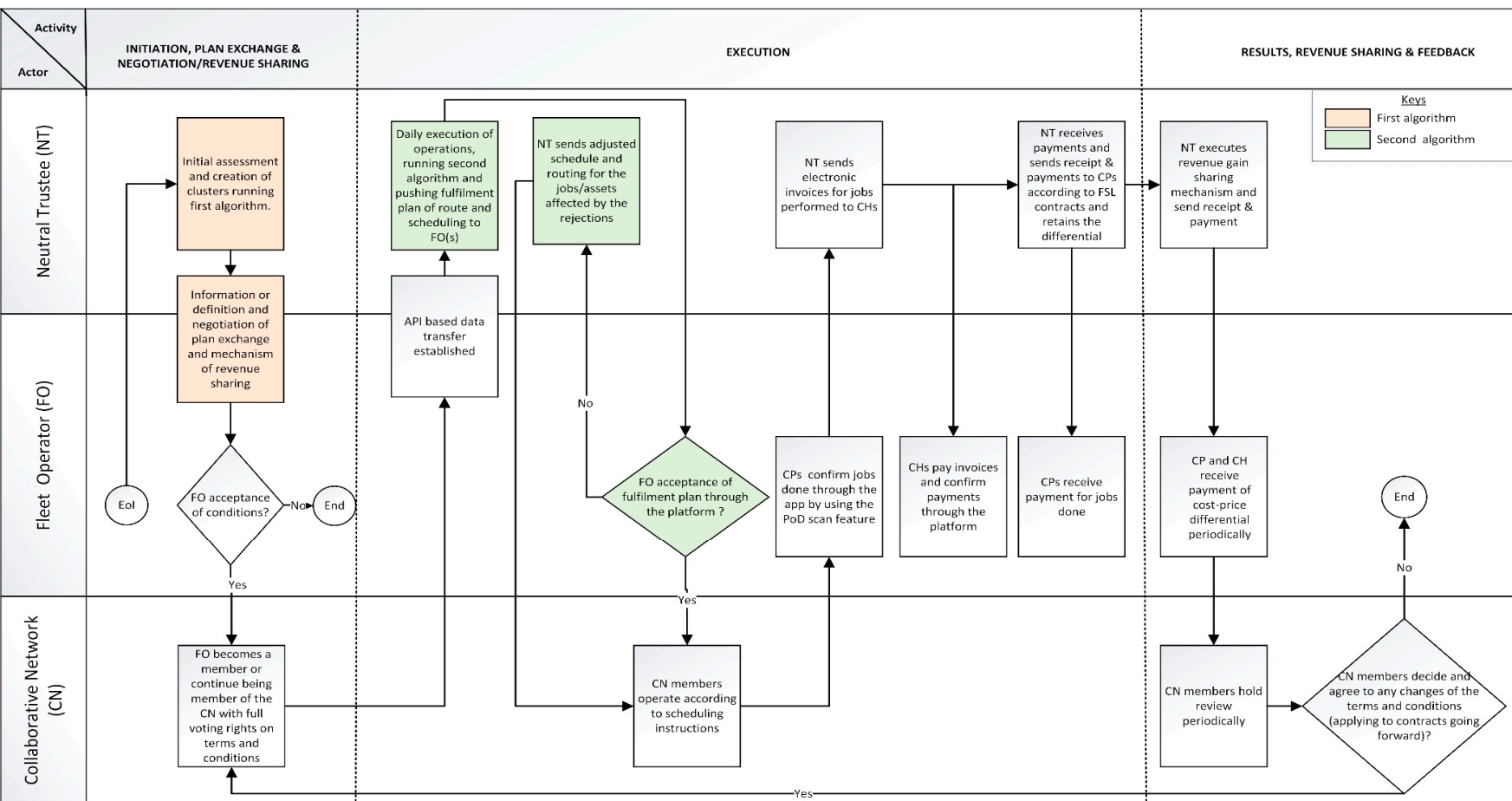

**Figure 7.** Updated collaboration process workflow integration between the gain-sharing business model and specialist planning algorithms for logistics collaboration from [1].

## 6. UK Case Study

### 6.1. Methods

Data from a major UK building supplier specialising in bricks and operating full truck-loads (FTL), hereafter referred to as Freight Operator X for confidentiality purposes, was provided in order for us to evaluate the potential benefits of the FSLP. This dataset comprised a full set of vehicles and orders from each of two depots.

In order to perform the evaluation, the following steps were performed.

1. To simulate the upload of the other fleets' data, anonymised data from 400 realistic HGV fleets were sampled from a commercially owned historic database of fleet data across the UK. This represented a scenario where a modest ~0.5% of operators used the FSLP.
2. A Fleet X depot was arbitrarily chosen, and fleet X data for that depot was uploaded to the FSLP.
3. The FSLP algorithms, as described in Sections 4.1 and 4.2 were then activated, resulting in fleet X becoming grouped with another operator, who we will refer to as 'Freight Operator B'. The FSLP automatically generated a collaborative plan for fleets X and B, and then provided the mileage, costs and other Key Performance Indicators (KPIs) of that plan.
4. In addition (via a special setting the FSLP back-end), the commercial fleet planning solver used in Step 3 (and referred to in Figure 5) was run to establish the corresponding mileage and costs in the case where fleets B and X worked alone.
5. Finally, Steps 1–4 were repeated 100 times in order to obtain robust results across a range of scenarios.

Once fleet B is chosen by the FSLP, the primary sharing method used by the collaborative optimization algorithm begins with a 'morning transfer' shuttling process. Here, for example, a vehicle of Freight Operator X takes some of Freight Operator X's jobs to Freight Operator B's depot and returns with some of Freight Operator B's jobs to deposit them at Freight Operator X's depot. The selected jobs are those that are more conveniently delivered from the other fleet's depot.

Finally, it should be noted that the FTL is challenging for freight share scenarios. Normally, a significant portion of FSL benefits arise from favourable alignments in the locations of the two freight operators' customers; but such alignments cannot be exploited in an FTL setting. Instead, benefits in the FTL setting are confined to efficiency savings in the 'morning transfer' process. These efficiencies arise from the fact that the shuttle trips typically carry an FTL each way, and reduce the overall mileage needed.

### 6.2. Efficiencies Achieved

From the scenarios tested, approximately one in four Freight Operator X datasets led to significant benefits when using FSL, when compared with the 'no sharing' scenario. It should be noted that we would expect a greater frequency of benefits for many other fleets, but this result partly reflects the challenging nature of the FTL setting. It is also worth noting that the heuristics that influence finding a matching fleet (share factor and affinity score, as described in Section 4) were not tuned to FTL scenarios.

Nevertheless, overall findings showed that cost and mileage savings varied between 2%–12% of the total costs and mileage that Freight Operator X and Freight Operator B would have otherwise incurred if operating individually. Additionally, applying the FSL gain-share business model and assuming a 5% profit margin, and a 1% cut for the operation of the FSLP itself, the increases in profit vary from 50% to 300%.

The following image, Figures 8 and 9, show the results of one of the scenarios where Freight Operator B collaborates with Freight Operator X's Depot 1 and Depot 2, respectively. The blue and red squares represent Freight Operator X's and Freight Operator B's jobs, respectively, while the yellow diamond with the coloured borders represent the location of each of their depots. A border of

the opposite colour implies the algorithm considers it more efficient for the other fleet to fulfil that specific job.

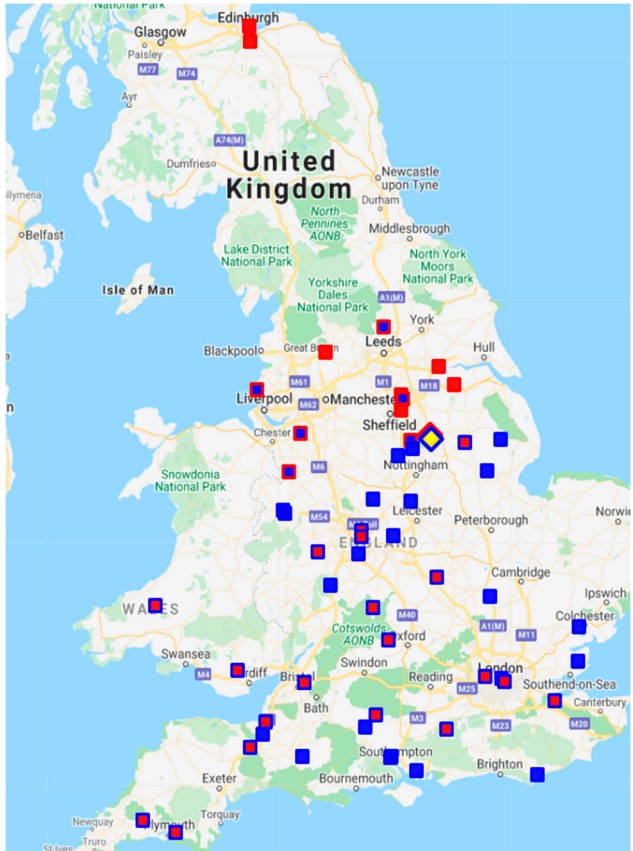

**Figure 8.** Depot 1.

As can be observed in Figure 8, Freight Operator B's depot is located slightly to the North of Freight Operator X's Depot 1, and so Freight Operator B takes on Depot 1's jobs that are closer to it. Altogether, Freight Operator B does 6 (23%) of Depot 1's jobs (blue squares with red border). Meanwhile, Freight Operator X handles 21 of Freight Operator B's jobs (of 26), located further south and central. The 'morning transfer' redistribution of these 27 loads between depots is accomplished by 10 Freight Operator X trucks and 11 Freight Operator B trucks between 08:00 and 09:00.

It should be noted that, for confidentiality purposes, the locations of the freight operators' depots and jobs shown in Figures 8 and 9 have been randomly moved within a 1 km radius of the real locations.

Table 8 provides a summary of the total costs and mileage incurred when both freight operators run individually and when collaborating, which leads to 7% and 6% reduction in mileage and costs, respectively.

**Table 8.** Freight Operator B and Freight Operator X, Depot 1 Collaboration.

| | Mileage (km) | | Costs (£) | |
|---|---|---|---|---|
| | **No Sharing** | **Collaboration** | **No Sharing** | **Collaboration** |
| **Freight Operator X, Depot 1** | 8920 | 16,050 | £103,291 | £185,190 |
| **Freight Operator B** | 10,753 | 2045 | £124,231 | £24,603 |
| **Collaboration leg: 'morning transfer'** | N/A | 168 | N/A | £3396 |
| **Total** | 19,673 | 18,263 | £227,522 | £213,189 |
| **Savings** | 1410 | | £14,333 | |
| **% Savings** | 7% | | 6% | |

As can be observed in Figure 9, Freight Operator B's depot is just south of Lancaster, and is hence better placed than Freight Operator X to handle the cluster of Claughton jobs in the Welsh border and Merseyside region, and therefore takes on 14 of Freight Operator X's jobs. Meanwhile, Freight Operator X handles 7 of Freight Operator B's jobs (out of 20), including a cluster located in the Leeds area, which is closer to Freight Operator X's Depot 2. The 'morning transfer' redistribution of these 21 loads between depots is accomplished by 7 of Freight Operator X's trucks and 7 of Freight Operator B's trucks between 08:00 and 09:00.

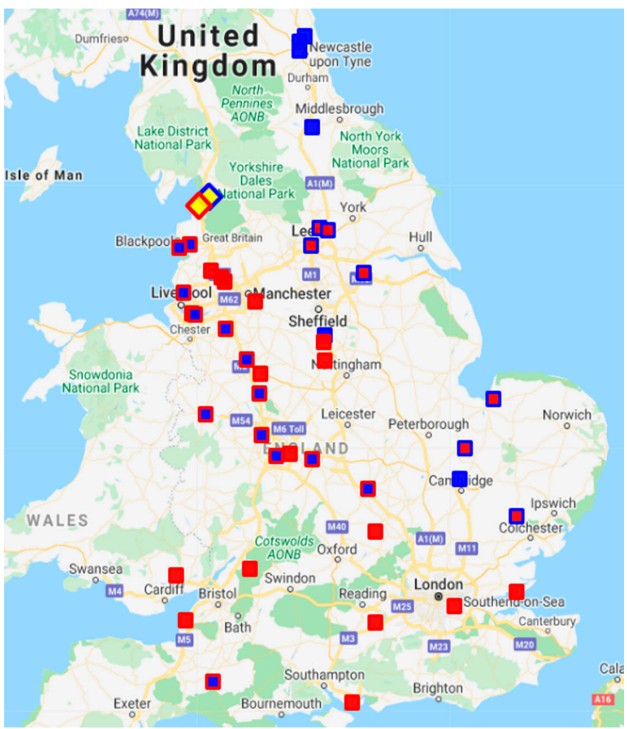

**Figure 9.** Depot 2.

A summary of mileage and costs incurred individually and when collaborating is included in Table 9. In this case, due to the distribution of both operators' jobs and the location of their depots, collaboration leads to a 12% and 11% reduction in mileage and jobs, respectively.

**Table 9.** Freight Operator B and Freight Operator X Depot 2 Collaboration.

| | Mileage (km) | | Costs (£) | |
|---|---|---|---|---|
| | No Sharing | Collaboration | No Sharing | Collaboration |
| **Freight Operator X Depot 2** | 7156 | 5339 | £82,600 | £61,357 |
| **Freight Operator B** | 8842 | 8435 | £101,814 | £98,925 |
| **Collaboration leg: 'morning transfer'** | N/A | 308 | N/A | £4670 |
| **Total** | 15,998 | 14,082 | £184,414 | £164,952 |
| **Savings** | 1916 | | £19,462 | |
| **% Savings** | 12% | | 11% | |

The benefits summarized in Tables 8 and 9 can be regarded as demonstrating the superiority of the Collaborative Optimization algorithm (in the 'Collaboration' columns) when contrasted with the current approach (in the 'No sharing' columns), in which each fleet is optimized individually using only its own resources to service its own tasks.

*6.3. Wider Socio-Economic and Environmental Benefits*

As per the validation described earlier on in this paper, the algorithm results, shown in Tables 8 and 9, have been utilised to quantify the annual change in the externalities associated with logistics operators' activities. These offer an understanding of the level of benefits that can be expected if these particular Freight Operators, X and B, were to collaborate for an entire year during similar type of operations (Tables 10 and 11).

**Table 10.** Annual mileage and emissions savings through the different collaboration scenarios.

| Collaboration Arrangement | Emissions Saved | | | |
|---|---|---|---|---|
| | $CO_2$ (tonnes) | PM (Kgs) | NOx (Kgs) | HC (Kgs) |
| **Freight Operator X Depot 1** | 212 | 5 | 559 | 5 |
| **Freight Operator X Depot 2** | 288 | 7 | 759 | 7 |

**Table 11.** Annual wider economic costs savings through the different collaboration scenarios.

| Collaboration Arrangement | Other Wider Economic Cost Savings | | |
|---|---|---|---|
| | Congestion (k£) | Infrastructure (k£) | Accidents (k£) |
| **Freight Operator X, Depot 1** | 103 | 62 | 2 |
| **Freight Operator X, Depot 2** | 140 | 84 | 2 |

As can be observed and in relation to Tables 8 and 9, the collaboration between Freight Operator B and Freight Operator X's Depot 2 leads to wider economic cost savings. This is due to a more efficient distribution being able to be achieved in this scenario due to the location of the jobs, as well as the depot being more favourable located.

**7. Conclusions**

Collaboration in the logistics industry has the capability to generate significant benefits for society, the environment and the economy, as it already has in many other sectors that are ahead with regards to collaboration. In this paper, we have contributed additional evidence and techniques in relation to logistics collaboration, particularly with regard to demonstrating business benefits in the context of real-world data. Part of the evidence for the benefits of collaboration provided herein includes a summary and further elaboration of recent studies [1,2] that explored asset sharing using data from

116 real fleets. The latter study showed that 25 pairs of fleets (hence involving over 40% of these fleets) could yield significant benefit in theory from an FSLP-style approach. The main additional evidence presented in this paper was from the two UK case studies in Section 6, in which we directly applied the FSLP prototype to data from two potential FSL-participant fleets, again showing significant achievable benefits in both cases. Overall, using the models and algorithms developed, a strong business case for collaboration for operators through asset sharing is evident. The business case is especially compelling for companies whose base of operations is within a region that is dense with industry, since this greatly increases the number of potential collaborators, and greater efficiencies are achieved.

However, even when the theory shows evidence of benefits for industry and society using collaboration in the logistics industry, the reality is that the market does not appear to be willing to embrace such a change. Logistics operators navigate a highly competitive, low-profit margin market and can often not cope with the risks associated with new technologies. This justifies their reluctance to any disruption to their operating models which have been functioning during the last decades. Their reluctance to share potentially commercially sensitive data has led to limitations in the number of model runs which have been undertaken. However, even in the more challenging FTL case study analysed herein, results are quite promising with regards to the operating cost savings and wider economic benefits achieved. Given that a significant portion of FSL efficiencies arise from favourable alignments in the locations of freight operators' customers, which cannot be exploited in an FTL setting, further benefits and efficiencies are expected from multi-drop operations. In this context, it is also worth noting that the Collaborative Optimization algorithm of Section 4.2, despite having demonstrated superiority to the incumbent approach, could well be improved or replaced. For example, alternative collaborative algorithms could be designed to exploit the features of particular fleet combinations, such as combining multi-drop and FTL fleets.

Hence, further empirical case studies are recommended to verify these findings and quantify the degree to which the business benefits depend on the regional industry demographics. This should be done in parallel to the dissemination of the benefits collaboration could offer to logistics operators, where the virtualisation tool created as part of the FSL project is utilised to test and understand the impact collaboration may have on each individual business.

The FSL project was completed in April 2020, having refined the key algorithms and taken the development of the FSLP to TRL 6. Real-world trials are then expected to follow during 2020, with a view to commercialising the FSLP thereafter. The impact of this on the traditional businesses of the logistics industry is likely transformational and hugely beneficial to the environment, the economy and society. It will provide an important boost for accelerated developments leading to the paradigm of the Physical Internet.

**Supplementary Materials:** The proposed business model is available online at https://youtu.be/AxMDnbLQes4.

**Author Contributions:** Conceptualization and analysis, A.V.; algorithms, D.C.; investigation, A.V. and C.F.; economic analysis, C.F. writing—original draft preparation, A.V., C.F. and D.C.; writing—review and editing, A.V., C.F. and D.C. All authors have read and agreed to the published version of the manuscript.

**Funding:** This research has been partially funded by Innovate UK.

**Acknowledgments:** We acknowledge the funding received by Innovate UK, project number 103890, titled 'FreightShare Lab' (FSL). We thank our colleagues from the FSL consortium: TrakM8 (formerly Route Monkey), DVV Media International, Connected Places Catapult Ltd. (formerly Transport Systems Catapult Ltd.) and Herriot-Watt University, who provided valuable insight and expertise that greatly supported this research. This paper and the research behind it would not have been possible without the exceptional support and guidance of Andrew Trail. His enthusiasm, knowledge and expertise were of invaluable insight for us and we will be always grateful for it.

**Conflicts of Interest:** The authors declare no conflict of interest.

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
