# Peer review of "Towards Sustainable Collaborative Logistics Using Specialist Planning Algorithms and a Gain-Sharing Business Model: A UK Case Study"

_sustainability, doi:10.3390/su12166627_

Round 1
Reviewer 1 Report
The introduction doesn't make clear that it's focussed on carrier/logistics service providers as opposed to shippers, nor does it say about what type of carrier, nor what type of products it is suitable for (are car transporters, concrete mixers suitable - obviously not relevant). It is also unclear what is meant by Contract Holder or Contract Performer. There are a few typos to be corrected but generally speaking (from what I've read so far) I like what they've been doing.
Author Response
Response to Reviewer 1 Comments
Dear reviewer,
First of all, many thanks for your comments and suggestions to improve the paper.
Point 1:The introduction doesn't make clear that it's focussed on carrier/logistics service providers as opposed to shippers, nor does it say about what type of carrier, nor what type of products it is suitable for (are car transporters, concrete mixers suitable - obviously not relevant). It is also unclear what is meant by Contract Holder or Contract Performer.
Response 1:
The introduction has been reinforced clearly indicating the focus of the business model on logistics services providers. It has been clarified the definitions of Contract Holder and Contract Performer in the introduction.
Point 2: There are a few typos to be corrected but generally speaking (from what I've read so far) I like what they've been doing.
Response 2:
The typos have been corrected and can be checked on the track changes.
Reviewer 2 Report
The paper at hand introduces a platform for collaborative logistics and explains the underlying business model. As the authors clearly point out in the introduction, research in this area is relevant as real-world attempts at establishing collaborative logistics schemes often fail in spite of potential benefits to sustainable transport. The proposed platform both proposes partners according to operational compatibility and supports planning and optimisation.
While the paper describes the potential use of the platform well, I miss a critical consideration of two primary barriers, which in my eyes need to be overcome to enable collaborative logistics on the long-term:
- Ensuring that joint planning can be realised without the partners giving up data that they may not want to give up to keep a competitive advantage or may not be allowed to give up to uphold antitrust-regulations, as mentioned, for example, by the references 16 and 23-26 cited by the authors
- Gain and cost sharing according to game theoretical insights, as proposed by, for example, the references 11, 16, 24 and 28 cited by the authors
The authors laudably mention these barriers and also state strategies to overcome them. However, in the remainder of the paper, the requirements to implement these strategies and their exact implementation are neglected. To ensure this consideration, I propose numbering the barriers given in Table 1 and referencing them systematically in the further paper, thereby systematically considering the presented platform in their light.
Minor remarks:
- Please carefully render all figures, as they are somewhat fuzzy and hard to read. A particular example is Figure 5.
- Table 3 still contains some referencing and formatting errors.
Author Response
Response to Reviewer 2 Comments
Dear reviewer,
First of all, many thanks for your comments and suggestions to improve the paper.
Point 1: The paper at hand introduces a platform for collaborative logistics and explains the underlying business model. As the authors clearly point out in the introduction, research in this area is relevant as real-world attempts at establishing collaborative logistics schemes often fail in spite of potential benefits to sustainable transport. The proposed platform both proposes partners according to operational compatibility and supports planning and optimisation.
While the paper describes the potential use of the platform well, I miss a critical consideration of two primary barriers, which in my eyes need to be overcome to enable collaborative logistics on the long-term:
- Ensuring that joint planning can be realised without the partners giving up data that they may not want to give up to keep a competitive advantage or may not be allowed to give up to uphold antitrust-regulations, as mentioned, for example, by the references 16 and 23-26 cited by the authors
- Gain and cost sharing according to game theoretical insights, as proposed by, for example, the references 11, 16, 24 and 28 cited by the authors
The authors laudably mention these barriers and also state strategies to overcome them. However, in the remainder of the paper, the requirements to implement these strategies and their exact implementation are neglected. To ensure this consideration, I propose numbering the barriers given in Table 1 and referencing them systematically in the further paper, thereby systematically considering the presented platform in their light.
Response 1:
We have followed your suggestions and included numbering to the barriers of Table 1 and these have been referenced in the rest of the paper. The same exercise has been done for Table 2 to give consistence to the paper.
Point 2: Please carefully render all figures, as they are somewhat fuzzy and hard to read. A particular example is Figure 5
Response 2: Figures 2 and Figure 5 (Now Figure 6) have increased the quality for the reader to be able to read without difficulty.
Point 3: Table 3 still contains some referencing and formatting errors.
Response 3: The referencing and format has been corrected in Table 3. Since the tracker changes does not track changes into the Table 3, the changes done are reflected here:
“The definition of the key activities in a collaborative process has been proposed in [10], [46] based on previous ideas from [5], [19], [24], [48]–[51]. These activities are: 1) Initiation, 2) Plan Exchange, 3) Negotiation/Revenue Sharing, 4) Execution, 5) Performance Measurement & Revenue Sharing Execution, and 6) Feedback/ Improvement."
Reviewer 3 Report
In general, the paper is ok. However, some improvements are necessary. Therefore, p. 9/25 - reference not found. So, please, insert reference.
Figure 5 is simply unreadable, unclear, and must be improved.
If Authors are writing about two algorithms, so they must be presented in diagram forms So far, the 2 algorithms are included in Figure 2 only as tasks in workflows.
Author Response
Response to Reviewer 3 Comments
Dear reviewer,
First of all, many thanks for your comments and suggestions to improve the paper.
Point 1: In general, the paper is ok. However, some improvements are necessary. Therefore, p. 9/25 - reference not found. So, please, insert reference.
Response 1:
Since the tracker changes does not track changes into the Table3, the changes done are reflected here:
“The definition of the key activities in a collaborative process has been proposed in [10], [46] based on previous ideas from [5], [19], [24], [48]–[51]. These activities are: 1) Initiation, 2) Plan Exchange, 3) Negotiation/Revenue Sharing, 4) Execution, 5) Performance Measurement & Revenue Sharing Execution, and 6) Feedback/ Improvement.
Point 2: Figure 5 is simply unreadable, unclear, and must be improved.
Response 2: Figures 5 (Now Figure 6) has increased the quality for the reader to be able to read without difficulty.
Point 3: If Authors are writing about two algorithms, so they must be presented in diagram forms So far, the 2 algorithms are included in Figure 2 only as tasks in workflows.
Response 3: As requested, the two key algorithms – described verbally in sections 4.1 and 4.2 respectively, are now also presented in detailed pseudocode in Figures 3 and 4.
Reviewer 4 Report
"Towards sustainable collaborative logistics using specialist planning algorithms and a gain-sharing business model: a UK case study" is an article providing a detailed discussion of the issues surrounding the use of the FreightShare Lab Platform (FSLP) and its embedded business model aiming to facilitate collaboration in freight logistics. In my opinion, the idea is very interesting and innovative. The authors focus on not only on theoretical aspects but also on the application of the proposed approach in practice, make the article of interest to both researchers and practitioners alike.
I believe that the article is at a fairly early stage of development. The originality and scientific level of the article is low. In the paper, authors have not presented the research problem, the scientific aim and research methods. The paper reads more like a report than an academic research paper. In my opinion, the paper requires more work regarding the methodology of research.
1) The article is slightly undermined by the lack of a clearly specified statement of the intentions of the author. The article's aim is not clearly defined in the abstract and introduction sections.
2) In the paper lack of "Methods section". The research method is weak. There is no basic theoretical model of research, its formalization. There is no synthetic description of the research methodology used. The model of research should be created.
3) Figure 2. and Figure 5. are illegible.
4) The novelty of this paper is not clearly presented. Finally, briefly mention the main contributions of the paper both at the end of the introduction and in conclusion.
Author Response
Response to Reviewer 4 Comments
Dear reviewer,
First of all, many thanks for your comments and suggestions to improve the paper.
Point 1: "Towards sustainable collaborative logistics using specialist planning algorithms and a gain-sharing business model: a UK case study" is an article providing a detailed discussion of the issues surrounding the use of the FreightShare Lab Platform (FSLP) and its embedded business model aiming to facilitate collaboration in freight logistics. In my opinion, the idea is very interesting and innovative. The authors focus on not only on theoretical aspects but also on the application of the proposed approach in practice, make the article of interest to both researchers and practitioners alike.
I believe that the article is at a fairly early stage of development. The originality and scientific level of the article is low. In the paper, authors have not presented the research problem, the scientific aim and research methods. The paper reads more like a report than an academic research paper. In my opinion, the paper requires more work regarding the methodology of research.
Response 1: The reviewer is correct in saying that the research problem, scientific aim and research methods are not clear. The main point that we would make in reply is to emphasise that this paper is a case study paper rather than a pure research paper. Nevertheless, this comment has led us to improve the paper in the following ways:
- We have now added a Methods section as subsection 6.1, the first part of the Case Study (section 6) - this section includes additional material which clarifies the experimental background and setup for the case study;
- We have adjusted the abstract to note that the aim of the work was to test the key FSLP algorithms on real-world data (as also now mentioned in the Methods section ).
Point 2: The article is slightly undermined by the lack of a clearly specified statement of the intentions of the author. The article's aim is not clearly defined in the abstract and introduction sections.
Response 2: The Abstract, Introduction and Conclusions have been improved to give clarity about the goals of the paper.
Point 3: In the paper lack of "Methods section". The research method is weak. There is no basic theoretical model of research, its formalization. There is no synthetic description of the research methodology used. The model of research should be created.
Response 3: As discussed in Response 1 above, we have now added a Methods section.
Point 4: Figure 2. and Figure 5. are illegible.
Response 5: Figures 2 and Figure 5 (Now Figure 6) have increased the quality for the reader to be able to read without difficulty.
Point 5: The novelty of this paper is not clearly presented. Finally, briefly mention the main contributions of the paper both at the end of the introduction and in conclusion.
Response 5: The main contributions of the paper have been summarised on the introduction and conclusions as you kindly suggested.
Round 2
Reviewer 4 Report
The authors have improved the paper. They have taken into account some suggested suggestions mentioned in the review. The article is still at a fairly low scientific level. However, the article may be published and constitute an introduction to further, more in-depth research in the field under consideration.
Author Response
Dear Reviewer,
We really appreciate your suggestions and comments.
Point 1: The authors have improved the paper. They have taken into account some suggested suggestions mentioned in the review. The article is still at a fairly low scientific level. However, the article may be published and constitute an introduction to further, more in-depth research in the field under consideration.
Response 1:
On the second round, we have improved the introduction and conclusion and the changes included can be seen with the track changes manuscript.
The introduction has been reinforced clearly indicating the contribution in the third paragraph of the introduction section from lines 38 to 46.
We also recognise and clarify the fact that alternative collaborative algorithms could be designed to further improve outcomes, and some explanation of that has been added to section 7 (conclusions) at the end of paragraph 2, lines 673 to 677.
We thank the reviewer for this comment, which has led us to add further text to the paper which clarifies, brings together, and underlines the evidence provided in the paper. This is done via an additional 10 lines in the first paragraph of the Conclusions (section 7), lines 650 to 658. In short, the new text summarises and synthesises the evidence presented in sections 4.4 (Validation) and 6 (UK Case Study).
